# CSF1R regulates the dendritic cell pool size in adult mice via embryo-derived tissue-resident macrophages

Gulce Itir Percin[1,2], Jiri Eitler[1], Andrea Kranz [3], Jun Fu[3], Jeffrey W. Pollard [4,5], Ronald Naumann[6] & Claudia Waskow[1,2,7,8]

Regulatory mechanisms controlling the pool size of spleen dendritic cells (DC) remain incompletely understood. DCs are continuously replenished from hematopoietic stem cells, and FLT3-mediated signals cell-intrinsically regulate homeostatic expansion of spleen DCs. Here we show that combining FLT3 and CSF1R-deficiencies results in specific and complete abrogation of spleen DCs in vivo. Spatiotemporally controlled CSF1R depletion reveals a cell-extrinsic and non-hematopoietic mechanism for DC pool size regulation. Lack of CSF1R-mediated signals impedes the differentiation of spleen macrophages of embryonic origin, and the resulted macrophage depletion during development or in adult mice results in loss of DCs. Moreover, embryo-derived macrophages are important for the physiologic regeneration of DC after activation-induced depletion in situ. In summary, we show that the differentiation of DC and their regeneration relies on ontogenetically distinct spleen macrophages, thereby providing a novel regulatory principle that may also be important for the differentiation of other hematopoietic cell types.

[1] Regeneration in Hematopoiesis and Animal Models in Hematopoiesis, Institute for Immunology, Fetscherstr. 74, 01307 Dresden, Germany. [2] Regeneration in Hematopoiesis, Leibniz-Institute on Aging – Fritz-Lipmann—Institute (FLI), Beutenbergstrasse 11, 07745 Jena, Germany. [3] Genomics, Biotechnology Center, TU Dresden, BioInnovationsZentrum, Tatzberg 47-49, 01307 Dresden, Germany. [4] MRC and University of Edinburgh Centre for Reproductive Health, Queen's Medical Research Institute, 47 Little France Crescent, Edinburgh EH16 4TJ, UK. [5] Edinburgh Cancer Research UK Centre, MRC Institute of Genetics and Molecular Medicine, University of Edinburgh, Crewe Road S, Edinburgh EH16 4TJ Scotland, UK. [6] Max Planck Institute of Molecular Cell Biology and Genetics, Transgenic Core Facility, Pfotenhauerstr. 108, 01307 Dresden, Germany. [7] Department of Medicine III, Faculty of Medicine, TU Dresden, Fetscherstrasse 74, 01307 Dresden, Germany. [8] Faculty of Biological Sciences, Friedrich-Schiller University, Fürstengraben 1, 07743 Jena, Germany. These authors contributed equally: Gulce Itir Percin, Jiri Eitler.  Correspondence and requests for materials should be addressed to C.W. (email: Claudia.Waskow@leibniz-fli.de)

Dendritic cells (DCs) are key modulators of the immune system by presenting antigen not only for the initiation of antigen-specific adaptive immune responses but also for the induction of self-tolerance in the absence of activating signals. DCs are short-lived and therefore continuously replenished by the progeny of adult hematopoietic stem cells (HSCs)[1]. Owing to striking overlaps of functional and morphological characteristics compared to other cells of the mononuclear phagocyte system, significant efforts were made to characterize DC identity based on the isolation of lineage-restricted or committed precursor cells, lineage tracing, and transcription and growth factor requirements important for DC differentiation[2,3]. Despite these efforts, definite information on the differentiation path and/or growth factor requirements for DC generation in vivo remain incomplete.

Fetal liver kinase 2 ligand (FLK2L, FLT3L, FL) stands out in its effects on DC differentiation because it efficiently promotes the expansion of DCs and their precursors in vivo[4,5] and the differentiation of all DC subsets in vitro[6]. Consistently, lack of FL or its receptor FLT3 (FLK2, CD135) results in markedly reduced DC numbers in vivo[4,5]. However, in both cases a sizable DC population persists in the spleen, strongly suggesting that a signal of a hitherto unknown kind synergizes with FLT3-mediated effects to ensure efficient differentiation of DCs. Combined lack of Flt3 and Csf2rb (encoding for granulocyte macrophage colony-stimulating factor receptor (GM-CSFR), interleukin (IL)-3Rb, IL-5Rb)[4] or of Fl and Csf2 (encoding for GM-CSF)[7] failed to affect or only partially aggravated DC differentiation, respectively, leaving growth factor requirements for spleen DC differentiation unknown[3]. FLT3 and CSF1R (M-CSFR, CD115) are the defining "markers" for the prospective separation of DC progenitor cells in the bone marrow (BM)[4,8], and CSF1R expression is associated predominantly with the propensity for the differentiation into conventional DCs[4,9,10]. Mice carrying Csf1r-null alleles show normal DC differentiation and numbers in peripheral lymphoid organs but have an impaired generation of specific nonlymphoid tissue DCs in the epidermis[11], lamina propria[12], dermis, lung, and kidney[13], and such mice show defects in the differentiation of monocytes into inflammatory DCs[14], indicating the involvement of CSF1R-mediated signals in DC differentiation.

During embryonic development yolk sac-derived erythro-myeloid progenitors (EMP) differentiate into tissue-resident macrophages (TR-Mps) including the red-pulp macrophages (RP-Mps) in the spleen, which are maintained throughout life and which are very slowly replaced by adult HSC-derived progeny[15,16]. EMPs emerge at embryonic day 8.5 (E8.5) in the yolk sac and shortly after migrate to the fetal liver where they initialize adult-like hematopoiesis restricted to erythroid, megakaryocytic, and macrophage lineages[17]. Embryo-derived TR-Mps are the only EMP-derived cell types known to persist in many organs throughout adulthood[15,18,19]. Emergence of RP-Mps depends on Spi-C[20] and Myb[16] transcription factors and they differentiate through Runx+[21], Tie2+CSF1R+[15], and Kit+[22] cellular intermediates, potentially sequentially. However, growth factor receptor signals that are important for their differentiation remain largely unknown. Functionally, spleen RP-Mps have mainly been implicated in systemic iron homeostasis[23].

Here we show that the combined deficiency of FLT3 and CSF1R leads to the absence of DCs in the spleen of juvenile and adult mice. Contrasting the cell-intrinsic requirement for FLT3-mediated signals, CSF1R contributes to DC differentiation via a cell-extrinsic mechanism. Using a novel mouse tool that allows the lineage tracing of embryo-derived macrophages, we show that CSF1R-mediated signals are crucial for the generation of embryonic macrophages that persist in the spleen after birth, the RP-Mps. We show here that RP-Mps are required to establish the spleen DC pool in FLT3-deficient animals, assigning a novel role

to embryo-derived RP-Mps in regulating the pool size of onto-genetically distinct hematopoietic cells. Finally, RP-Mps are important regulators of steady-state hematopoiesis in the adult mouse by continuously supporting DC regeneration. This data links blood cell differentiation of adult HSCs to a cell type of a different ontogeny, providing another layer of complexity to the regulation of innate immune cell differentiation.

## Results

**FLT3 and CSF1R double null mice lack DCs.** To test whether lack of CSF1R aggravates paucity of DCs on a FLT3-deficient background, we generated $Flt3^{-/-}$[24] $Csf1r^{-/-}$[25] double mutant mice that were born to normal Mendelian frequencies. $Flt3^{-/-}$; $Csf1r^{-/-}$ mice recapitulated the phenotype of $Csf1r^{-/-}$ mice[25] and lacked teeth prompting us to analyze the mice 18–21 days after birth. Consistent with previous results, $Flt3$ single mutant mice showed a severe reduction in the frequency of DCs[4], whereas DC differentiation was independent of CSF1R-mediated signals[11] (Fig. 1a, Supplementary Fig. 1a). In contrast, a highly significant loss of DCs occurred in mice double deficient for $Flt3$ and $Csf1r$ compared to $Flt3$-null mice alone. Analysis of large cohorts of mice confirmed the significant reduction of DCs in $Flt3^{-/-}$;$Csf1r^{-/-}$ mice (Fig. 1b). The reduction of DCs out-numbered loss of overall spleen cellularity as shown by the fold-change reduction of spleen leukocytes ($CD45^+$) versus DCs (Fig. 1b right, Supplementary Fig. 1b). $CD8^+$ or $CD11b^+$ DC subsets were affected equally (Supplementary Fig. 1c). The effect of $Flt3$ and $Csf1r$ double deficiency was specific for DCs since closely related macrophages (Fig. 1c, Supplementary Fig. 1d) and RP-Mps (Fig. 1d)[26] were not affected. Absence of spleen DCs was confirmed by immunohistology on spleen sections (Fig. 1e, Supplementary Fig. 1e). A potential contribution of genetic variations to the DC phenotype based on the use of outbred C57/BL/6J×C3Heb/FeJ mice was excluded by generating congenic $Flt3^{-/-}$;$Csf1r^{-/-}$ mice on the C57BL/6J genetic background that also showed the reduction of DCs (Supplementary Fig. 1f). We conclude that functional FLT3 and CSF1R receptors are pivotal for the generation of a normally sized DC pool in the spleen of 18–21-day-old mice.

**Normal numbers of DC progenitors in $Flt3^{-/-}$;$Csf1r^{-/-}$ mice.** DC progenitors are identified by the expression of FLT3 and CSF1R or CX3CR1-GFP on the surface of immature BM cells[4,8]. To determine whether CSF1R-mediated signals become crucial for the generation of committed DC precursors on a $Flt3$-null genetic background, we determined the fraction of CX3CR1-GFP-expressing cells within the lineage negative (Lin⁻) Sca-1⁻ compartment in the BM, because these cells identify macrophage dendritic cell precursors (MDPs)[8] (Fig. 1f, g, Supplementary Fig. 1g). MDPs were found to comparable frequencies in $Flt3^{-/-}$; $Csf1r^{-/-}$;$Cx3cr1$-$gfp^+$ and $Flt3^{-/-}$;$Csf1r^{+/-}$;$Cx3cr1$-$gfp^+$ control mice. Furthermore, frequencies and numbers of pre-cDC1 and pre-cDC2 in the BM and spleen were normal in $Flt3$ and $Csf1r$ double deficient mice (Supplementary Fig. 1h-j). Taken together, CSF1R signaling in $Flt3$-deficient mice is dispensable for the generation of BM and spleen DC progenitors but CSF1R-mediated signals are important for DC differentiation beyond the MDP and pre-cDC stages.

**Cell-extrinsic and non-hematopoietic mechanism.** To test whether CSF1R expression is cell-intrinsically required for the generation of DCs beyond the MDP stage, we used a conditional $Csf1r$ allele[27] and generated $Flt3^{-/-}$;$Csf1r^{F/-}$ mice combined with CD11c-Cre deleter mice that express the Cre-recombinase at late stages during DC differentiation[28] (Fig. 2a, Supplementary

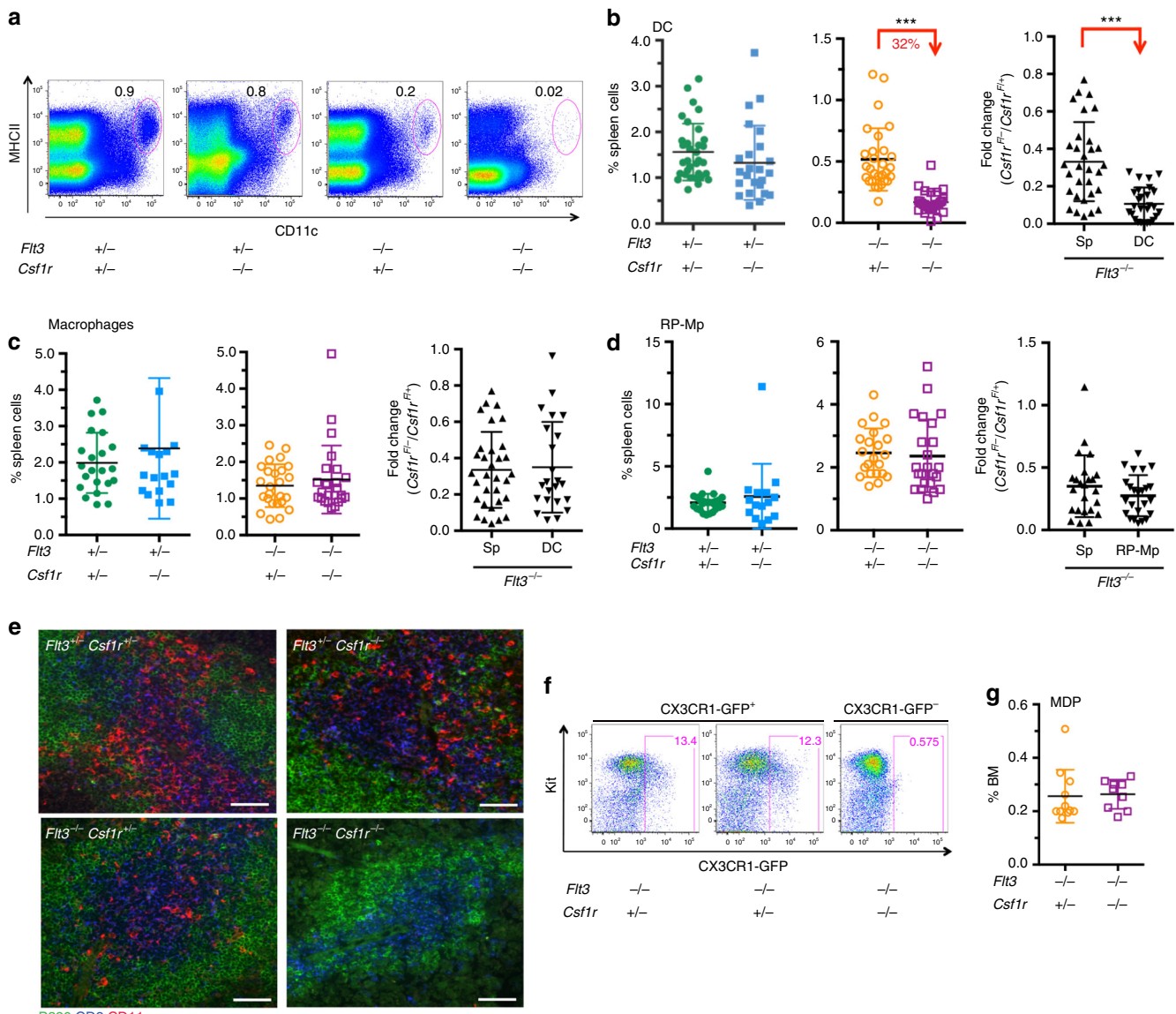

**Fig. 1** $Flt3^{-/-};Csf1r^{-/-}$ mice lack spleen DCs. **a** Flow cytometry of spleen cells from wild-type, $Flt3^{-/-}$, $Csf1r^{-/-}$, and $Flt3^{-/-};Csf1r^{-/-}$ mice. Numbers indicate frequencies of dendritic cells (DCs, CD11c$^{hi}$ MHCII$^{hi}$) within Dapi$^-$ cells. **b** Summary of DC frequencies (left, middle) in growth factor mutant mice. Right plot shows comparisons of fold changes between absolute leukocytes (CD45$^+$) and DCs from the spleens of wild-type and receptor-deficient mice to normalize for overall changes in cellularity. Absolute cell numbers are shown in Supplementary Fig. 1b. Two-sided $t$ test (left) and Mann–Whitney $U$ test (right) were performed. SD is shown. **c** Frequencies and fold-change comparison of spleen macrophages (Gr-1$^{lo/-}$ CD11b$^+$ F4/80$^{lo}$ SSC$^{lo}$) of wild-type and receptor-deficient mice as indicated. Gating is shown in Supplementary Fig. 1a. Two-sided $t$ test (left) and Mann–Whitney $U$ test (right) were performed. SD is shown. **d** Frequencies and fold-change comparison of spleen red-pulp macrophages (RP-Mps, Gr-1$^{lo/-}$ CD11b$^{lo}$ F4/80$^{hi}$ SSC$^{lo}$) of wild-type and receptor-deficient mice as indicated. Two-sided $t$ test (left) and Mann–Whitney $U$ test (right) were performed. SD is shown. **e** Immunohistology of spleen sections of 3-week-old wild-type and receptor-deficient mice as indicated. Sections were stained using specific antibodies recognizing B220 (green), CD3 (blue), and CD11c (red). ×20 objective was used for picture acquisition, scale bar corresponds to 50 μm. Pictures are representative of three mice analyzed for each genotype. **f** Dot plots show the expression of CX3CR1-GFP in Lin$^-$ (Lin = CD3, CD19, TER119, NK1.1, CD11b, CD11c, B220, Gr-1) Sca-1$^{lo/-}$ bone marrow hematopoietic progenitor cells in $Flt3^{-/-};Csfr1^{+/-};Cx3cr1\text{-}gfp^+$ or $Flt3^{-/-};Csf1r^{-/-};Cx3cr1\text{-}gfp^+$ mice. **g** Plot shows the quantification of macrophage dendritic cell progenitor (MDP) frequencies in the bone marrow as shown in **f**. Two-sided $t$ tests was performed and SD is shown

Fig. 2a). Efficient recombination was confirmed by molecular analysis (Supplementary Fig. 2b). $CD11c\text{-}Cre^+;Flt3^{-/-};Csf1r^{F/-}$ and $CD11c\text{-}Cre^+;Flt3^{-/-};Csf1r^{F/+}$ control mice contained comparable numbers of DCs, suggesting that the DC pool size in the spleen is independent of CSF1R-mediated signals in mature DCs (Fig. 2a, right).

To test whether CSF1R expression on any hematopoietic cell in the adult mouse is important for DC differentiation in the context of FLT3 deficiency, CSF1R was depleted on HSCs using $Vav\text{-}Cre$ deleter mice[29]. The $Csf1r^F$ allele was efficiently recombined in $Vav\text{-}Cre^+;Flt3^{-/-};Csf1r^{F/-}$ mice (Supplementary Fig. 2c). However, lack of CSF1R on hematopoietic cells of adult origin had no effect on DC numbers compared to controls as revealed by comparing the fold-change reduction between spleen cells and DCs (Fig. 2b, right), suggesting that the regulation of the DC pool size is independent of CSF1R-mediated signals in hematopoietic cells.

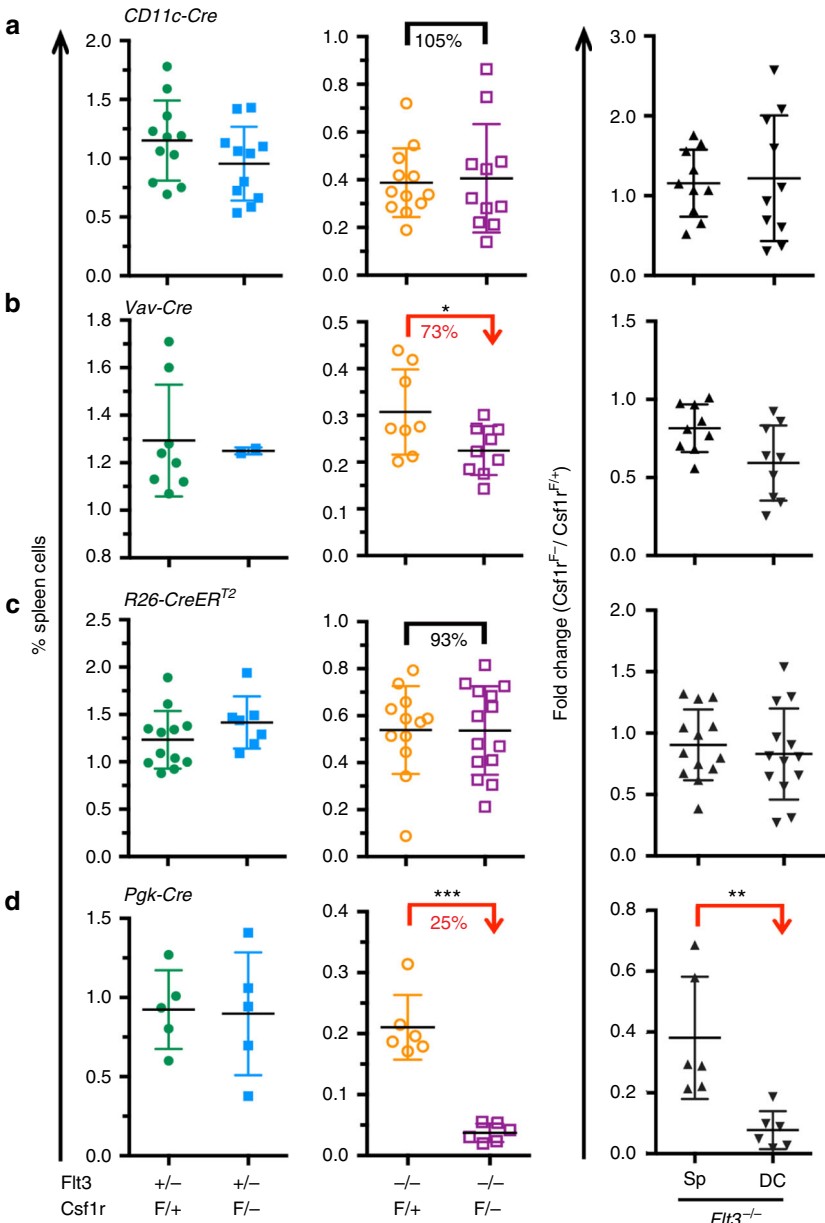

**Fig. 2** CSF1R expression on mature DCs, or on any hematopoietic cell type, or on any cell in adult mice is not required for DC generation. **a** DC frequencies (left, middle) and fold-change differences of leukocytes and DCs (right) in the spleen from *CD11c-Cre+;Flt3−/−;Csf1rF/−* and control mice at 3 weeks of age. **b** DC frequencies and fold-change differences of leukocytes and DCs in the spleen from *Vav-Cre+;Flt3−/−;Csf1rF/−* and control mice at 3 weeks of age. **c** DC frequencies and fold-change differences of leukocytes and DCs in the spleen from adult *R26-CreER^T2+;Flt3−/−;Csf1rF/−* mice. Mice were tamoxifen (TAM) induced 6–8 weeks earlier at 8–12 weeks of age. Scheme of the experimental outline is shown in Supplementary Fig. 2d. **d** DC frequencies and fold-change differences of leukocytes and DCs in the spleen from *Pgk-Cre+;Flt3−/−;Csf1rF/−* and control mice at 3 weeks of age. Two-sided *t* tests (left) and Mann–Whitney *U* tests (right) were performed in all figure parts. SD is shown throughout the figure

To test for a role of CSF1R expression on any cell in the adult mouse, *R26-CreER^T2+;Flt3−/−;Csf1rF/−* mice[30] were generated and CSF1R expression depleted by tamoxifen (TAM) induction in adult mice (Fig. 2c, scheme Supplementary Fig. 2d). CSF1R depletion was found efficient 1–2 weeks after induction on blood monocytes (Supplementary Fig. 2e, f) and 4–6 weeks later, at the time point of analysis, on BM precursor cells (Supplementary Fig. 2g) and on spleen macrophages (Supplementary Fig. 2h). Molecular analysis confirmed the efficient recombination of the *Csf1rF* allele in the BM and spleen (Supplementary Fig. 2i). This strategy ensured that DC differentiation took place in the absence of CSF1R expression for 4–6 weeks. Given that the replacement of the DC pool in the spleen takes 7–14 days[1], effects based on the

lack of CSF1R expression should become evident. The depletion of large peritoneal macrophages[31] but not spleen RP-Mps (Supplementary Fig. 2j, Fig. 1d) confirmed the success of this strategy. In contrast, DC numbers in the spleens of TAM-induced *R26-CreER^T2+;Flt3−/−;Csf1rF/−* mice were comparable to controls (Fig. 2c), suggesting that DC differentiation occurs independently of CSF1R expression in the entire adult organism.

Full functionality of the "floxed" *Csf1r* allele[27] was confirmed using *Pgk-Cre+* deleter mice that express the Cre-recombinase in oocytes through maternal transmission, ensuring ubiquitous and complete recombination of LoxP-flanked alleles[32] (Supplementary Fig. 2k). Spleens of *Pgk-Cre+;Flt3−/−;Csf1rF/−* mice lacked DCs (Fig. 2d), recapitulating the DC phenotype of constitutive

$Flt3^{-/-};Csf1r^{-/-}$ mice. Taken together, these results strongly suggest that CSF1R expression in the adult mouse is dispensable for normal DC differentiation pointing at a role for CSF1R-mediated signals during development.

### Fetal liver CD11b$^{lo}$ F4/80$^{hi}$ macrophages depend on CSF1R.

During development, EMP give rise to spleen CD11b$^{lo}$ F4/80$^{hi}$ RP-Mps that persist throughout life[15]. During that process, cellular intermediates express the CSF1 receptor[15,16,21]. In the yolk sac as well as throughout the body of E10.5 embryos, CSF1R is exclusively expressed within the CD45$^+$ fraction (Fig. 3a–c). Moreover, all CSF1R-positive cells co-express the macrophage marker F4/80 (Fig. 3b, c). To test for functional effects of CSF1R depletion on the formation of embryonic macrophages, mothers of $R26$-$CreER^{T2}$;$Csf1r^{F/-}$ mice were TAM-treated at E10.5 and

fetal liver cells analyzed 4 days later at E14.5 (scheme in Fig. 3d, top). The frequency of CD11b$^{lo}$ F4/80$^{hi}$ fetal liver embryonic macrophages was found significantly reduced compared to $R26$-$CreER^{T2+}$;$Csf1r^{F/+}$ controls (Fig. 3d, e), evidencing the dependency of embryonic macrophage differentiation on CSF1R-mediated signals. The effect was cell-type specific because CD11b$^+$ F4/80$^{lo}$ macrophages that originate from adult-type definitive HSCs were unaltered by the depletion of CSF1R (Fig. 3d, e). We conclude that CSF1R is expressed by CD11b$^{lo}$ F4/80$^{hi}$ fetal liver embryonic macrophages and/or their immediate progenitors and that they depend on CSF1R-mediated signals for their generation and/or maintenance.

### Spleen CD11b$^{lo}$ F4/80$^{hi}$ macrophages depend on CSF1R.

To test whether the depletion of CSF1R in the embryo has a direct

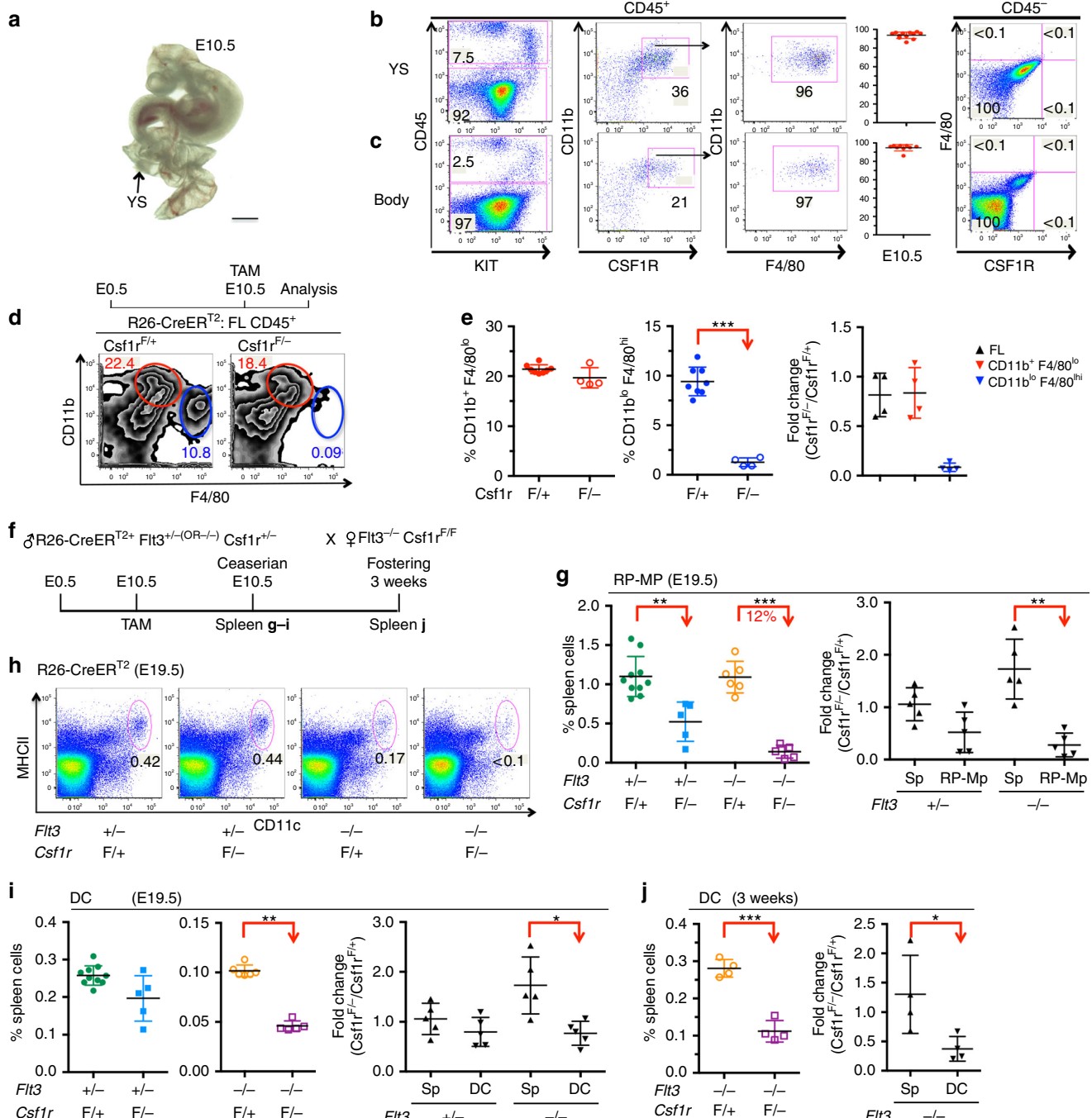

effect on the establishment of the DC pool, mothers of $R26$-$CreER^{T2+}$;$Flt3^{-/-}$;$Csf1r^{F/-}$ embryos were TAM-induced at E10.5 and analyzed at birth (E19.5) or at 3 weeks of age (scheme, Fig. 3f). Consistent with the phenotype of CD11b$^{lo}$ F4/80$^{hi}$ embryonic macrophages in the fetal liver (Fig. 3d, e), the population of spleen CD11b$^{lo}$ F4/80$^{hi}$ RP-Mps was significantly reduced irrespective of the genotype for $Flt3$ (Fig. 3g). We next tested for the effects on DC differentiation: In $R26$-$CreER^{T2+}$; $Flt3^{-/-}$;$Csf1r^{F/-}$ but not in control mice, spleen DCs were found severely reduced in newborn (E19.5, Fig. 3h, i) and 3-week-old mice (Fig. 3j). However, TAM-induced depletion of CSF1R expression in the embryo resulted in ablation of CSF1R expression on BM precursor cells at 3 weeks of age (Supplementary Fig. 2l). We conclude that the generation of CD11b$^{lo}$ F4/80$^{hi}$ embryo-derived TR-Mps in the spleen depends on CSF1R-mediated signals during development and abrogation of these signals results in lack of spleen DCs in FLT3-deficient mice. Thus this experimental setting underlined the importance for CSF1R expression during embryonic development but formally cannot exclude that CSF1R-mediated signals during postnatal stages contribute to the DC phenotype.

**Depletion of CSF1R in embryonic macrophages**. To directly test whether CSF1R-mediated signals are only important during the differentiation of embryonic macrophages, a novel lineage-tracing mouse model was generated by introducing the codon-improved Cre (iCre)-recombinase into the $Tnfrsf11a$ ($Rank$) gene locus (Fig. 4a, Supplementary Fig. 3a). Receptor activator of nuclear factor-κB (RANK) is an essential mediator for osteoclast development and diseases associated with mutations at this locus include familial expansile osteolysis and autosomal-recessive osteopetrosis, pointing at a role for the RANK ligand–RANK signaling pathway in the differentiation of embryo-derived macrophages. Lineage tracing using $Rank$-$iCre^+$;$eYFP^{wt/ki}$ mice revealed a specific label of embryo-derived spleen CD11b$^{lo}$ F4/80$^{hi}$ RP-Mps and Langerhans cells in the skin but no or very low labeling of hematopoietic cells derived from adult-type HSCs in the spleen, blood, BM, and skin (Fig. 4b, c, Supplementary Fig. 3b-d). Lack of labeling of short-lived hematopoietic cells in the blood, spleen, and skin and of long-term HSCs in the BM confirmed the absence of iCre expression in the stem and progenitor cell compartment after birth (Fig. 4c). Consistently, yolk sac embryonic macrophages but not fetal liver HSCs were efficiently labeled in $Rank$-$iCre^+$;$eYFP^{wt/ki}$ embryos (Fig. 4d, e), evidencing that the iCre recombinase is specifically expressed in embryo-derived TR-Mps. Spatiotemporally controlled depletion

of CSF1R in $Rank$-$iCre^+$;$Csf1r^{F/-}$ mice leads to the loss of CD11b$^{lo}$ F4/80$^{hi}$ RP-Mps but not of CD11b$^+$ F4/80$^{lo}$ Mp in the spleen of newborn mice (Fig. 4f). As a consequence, frequencies and numbers of DCs were reduced in $Rank$-$iCre^+$;$Flt3^{-/-}$; $Csf1r^{F/-}$ but not in $Rank$-$iCre^+$;$Flt3^{+/-}$;$Csf1r^{F/-}$ control mice (Fig. 4g). We conclude that CSF1R signaling is crucial for the generation of spleen embryo-derived RP-Mps during development that, in turn, are crucial for the establishment of the spleen DC pool in adult $Flt3^{-/-}$ mice.

**RP-Mps are crucial for the regeneration of spleen DCs**. To test whether CD11b$^{lo}$ F4/80$^{hi}$ RP-Mps are important for the maintenance of DC numbers in the spleen of adult FLT3-null mice, clodronate-loaded liposomes (Clod) were injected into $Flt3^{-/-}$ and wild-type mice (scheme, Fig. 5a). Because Clod treatment can result in inflammation, the depletion and regeneration of the affected cell types were closely monitored. Loss of blood monocytes 24 h after treatment (Fig. 5b, c) and of spleen macrophages, RP-Mps, and DCs (Fig. 5d) 1–2 weeks after injection confirmed the toxicity toward phagocytic cells. Two weeks later (4 weeks after Clod treatment), CD11b$^+$ F4/80$^{lo}$ spleen macrophages had recovered in all mice due replenishment from adult-type HSCs (Fig. 5e, left). In contrast, CD11b$^{lo}$ F4/80$^{hi}$ RP-Mps remained reduced irrespective of the $Flt3$ genotype (Fig. 5e, middle). DCs were significantly reduced in $Flt3^{-/-}$ but not in wild-type mice (Fig. 5e, right), suggesting that the importance for cell-extrinsic support for DC regeneration becomes evident in vivo in a situation where DCs and their progenitors carry a cell-intrinsic challenge, such as Flt3 deficiency.

Finally, we tested for the physiological relevance of DC/RP-Mp interaction in vivo. Spleen DCs regenerate after activation-induced depletion in vivo[33,34]. To determine whether DC regeneration in such a context depends on RP-Mps, we established a situation where DCs regenerate in a spleen that is devoid of RP-Mps (scheme, Fig. 6a). To this end, wild-type mice were treated with Clod and blood monocyte depletion was confirmed after 22 h (Fig. 6b). Blood monocytes (Fig. 6c), spleen CD11b$^+$ F4/80$^{lo}$ macrophages (Fig. 6d), and DCs (Fig. 6e) had recovered 4 weeks after Clod treatment to wild-type numbers. In contrast, RP-Mps were still reduced (Fig. 6f). At this time point, Clod-treated and control mice were injected with lipopolysaccharide (LPS) resulting in the activation of DCs in situ as determined by the increased expression of the co-stimulatory molecule CD86 (Fig. 6g). Following this activation, DC numbers expectedly decreased in all LPS-treated mice 2 days after activation (Fig. 6h). DC numbers quickly regenerated in mice

**Fig. 3** Embryo-derived spleen CD11b$^{lo}$ F4/80$^{hi}$ macrophages depend on CSF1R-mediated signals for their generation. **a** Photograph of E10.5 embryo with yolk sac (YS). Scale bar corresponds to 1 mm. **b**, **c** Dot plots show yolk sac (**b**) or body cells excluding head tissues (**c**) analyzed for the expression of CD45 and KIT (left). CD45-expressing cells are further resolved for the expression of CD11b and CSF1R (second left). CSF1R$^+$ cells co-express F4/80 (third left). Plots show the frequencies of F4/80-expressing cells within the CSF1R-positive population (second right). Each dot represents one mouse. CD45-negative cells are resolved for the expression of F4/80 and CSF1R (right). **d** Schematic outline of TAM-induced depletion of CSF1R expression in the embryo (top). Contour plots resolve CD45$^+$ fetal liver cells from E14.5 $R26$-$CreER^{T2+}$;$Csf1r^{F/-}$ and control embryos for the expression of CD11b and F4/80 (bottom). **e** Plots show the frequencies of CD11b$^+$ F4/80$^{lo}$ (left) or CD11b$^{lo}$ F4/80$^{hi}$ (middle) cells in the fetal liver of $R26$-$CreER^{T2+}$;$Csf1r^{F/-}$ and control embryos. Right plot shows fold-change differences of leukocyte and macrophage populations in the fetal livers. Data are summarized from three independent experiments. Two-sided $t$ test (left) and Mann–Whitney $U$ test (right) were performed. SD is shown. **f** Schematic outline of TAM-induced depletion of CSF1R expression in $R26$-$CreER^{T2+}$;$Flt3^{-/-}$;$Csf1r^{F/-}$ and control embryos. TAM was administered at E10.5. **g** Frequencies (left) and fold-change differences (right) of leukocytes and RP-Mps in the spleens of $R26$-$CreER^{T2+}$;$Flt3^{-/-}$;$Csf1r^{F/-}$ and control pups at E19.5. Data are summarized from four independent experiments. Two-sided $t$ test (left) and Mann–Whitney $U$ test (right) were performed. SD is shown. **h** Dot plots show DCs in $R26$-$CreER^{T2+}$;$Flt3^{-/-}$; $Csf1r^{F/-}$ and control mice at E19.5. Numbers indicate frequencies of dendritic cells within Dapi$^-$ cells. **i**, **j** Plots show DC frequencies and fold-change differences of leukocytes and DCs in the spleen of E19.5 pups (**i**) or at 3 weeks of age (**j**) in $R26$-$CreER^{T2+}$;$Flt3^{-/-}$;$Csf1r^{F/-}$ and control mice that were TAM-induced at E10.5. Data are summarized from three independent experiments. Two-sided $t$ tests (left) and Mann–Whitney $U$ tests (right) were performed. SD are shown

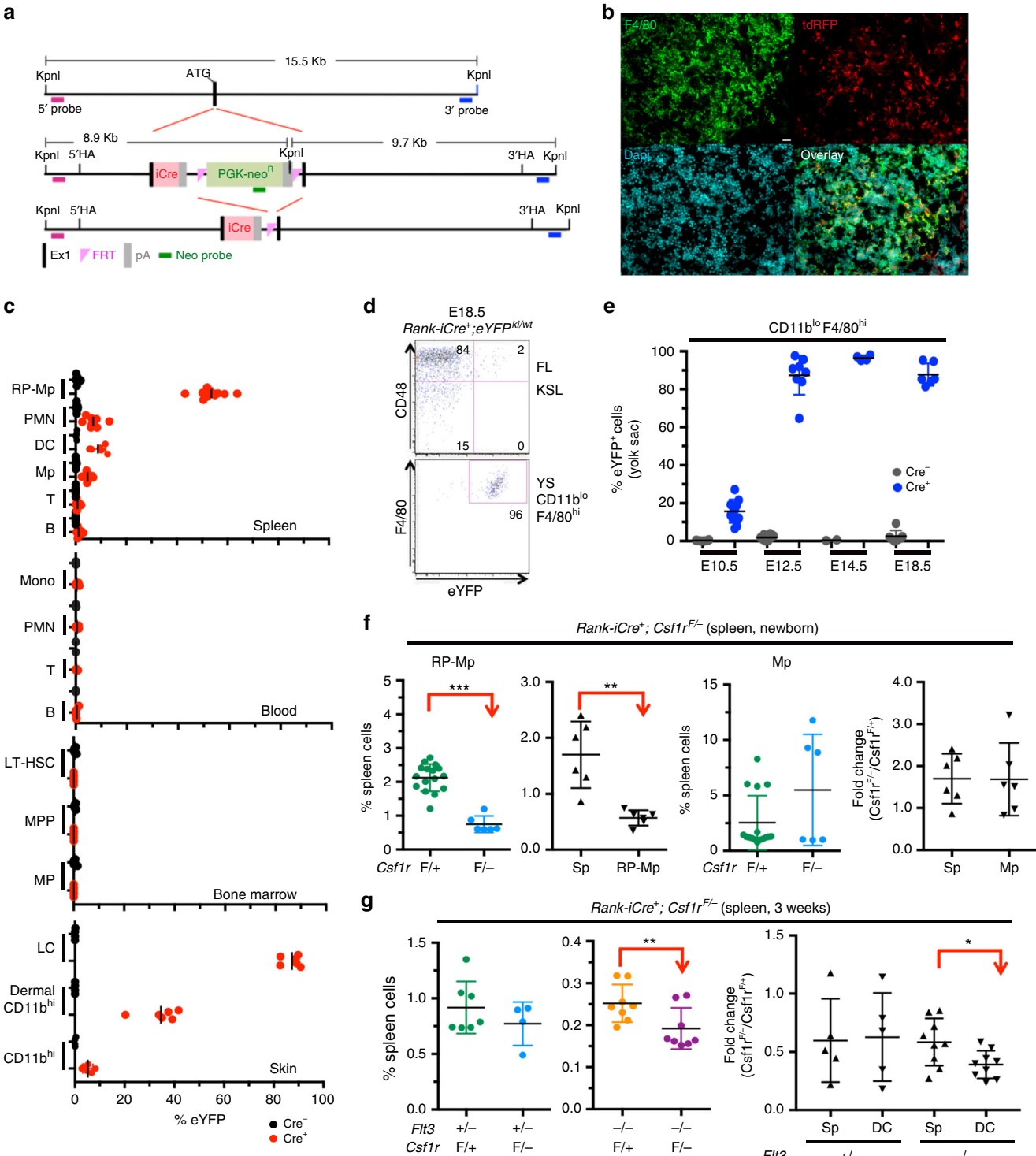

**Fig. 4** Tissue-specific depletion of CSF1R in RP-Mps and their progenitors results in loss of DCs. **a** Scheme of the generation of *Rank-iCre* mice (iCre = codon-improved Cre). The iCre-recombinase was knocked into exon 1 of the *Rank* gene. **b** Immunohistology on spleen sections from 3-week-old *Rank-iCre*⁺;*Td-rfp*⁺ mice, ×40 objective was used for picture acquisition, scale bar corresponds to 20 μm. **c** Lineage tracing of RANK-expressing cells in adult *Rank-iCre*⁺;*eYFP*^ki/wt mice. LT-HSC long-term HSCs, MPP multipotent progenitors, MP myeloid progenitors, LC Langerhans cells. SD is shown. Gatings are shown in Supplementary Fig. 1a and Supplementary Fig. 3b-d. **d** Dot plots show the labeling of fetal liver Kit⁺ Sca-1⁺ Lineage⁻ (KSL) hematopoietic stem and progenitor cells (top) and yolk sac CD11b⁺ F4/80^hi macrophages (bottom) from E18.5 *Rank-iCre*⁺;*eYFP*^ki/wt embryos. **e** Lineage tracing in *Rank-iCre*; *eYFP*^ki/wt embryos gating on CD11b^lo F4/80^hi embryonic macrophages in the yolk sac at the indicated time points. SD is shown. **f** Plots show the frequencies (left, second right) and fold changes (second left, right) of CD11b^lo F4/80^hi RP-Mps or CD11b⁺ F4/80^lo macrophages and spleen leukocytes from newborn *Rank-iCre*⁺;*Csf1r*^F/− and control mice. Two-sided *t* test (left and second right) and Mann–Whitney U test (right and second right) were performed. SD is shown. **g** Plots show frequencies and fold changes of DCs and spleen leukocytes from *Rank-iCre*⁺;*Flt3*⁻/⁻;*Csf1r*^F/− and control mice at 3 weeks of age. Two-sided *t* test (left) and Mann–Whitney *U* test (right) were performed. SD is shown

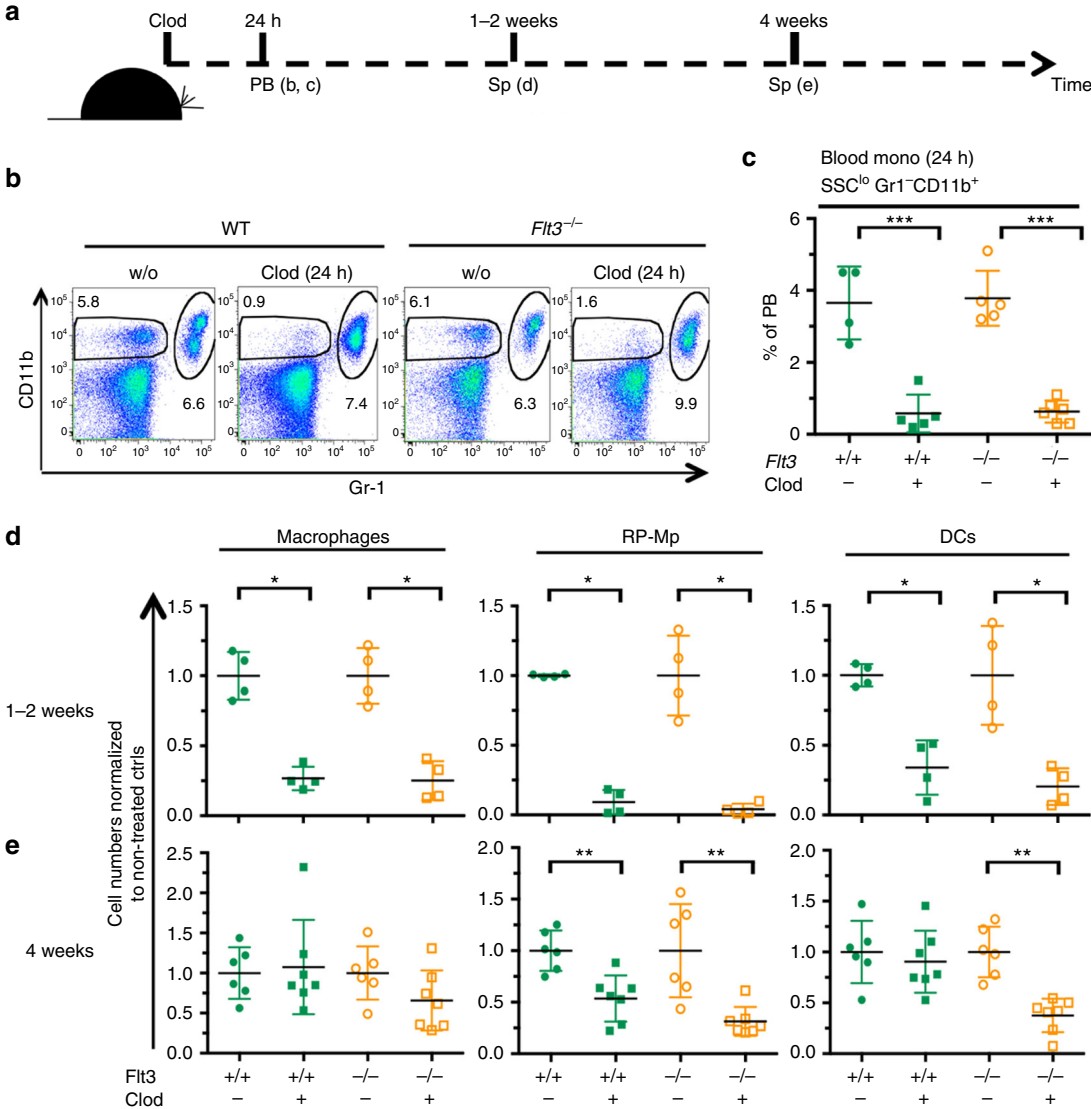

**Fig. 5** Depletion of spleen CD11b$^{lo}$ F4/80$^{hi}$ macrophages in adult FLT3-deficient mice results in loss of DCs. **a** Schematic outline of the experiment.
**b** Clodronate liposomes (Clod) were injected into *Flt3$^{-/-}$* and wild-type (WT) mice and the efficient depletion of blood monocytes (CD11b$^+$ Gr-1$^{lo}$) was analyzed 24 h later (left). **c** Plot shows the reduction of blood monocytes. Data are pooled from two independent experiments. SD is shown. **d** Cell numbers of spleen macrophages (left), RP-Mps (middle), and DCs (right) normalized to non-injected controls from the same experiments 1–2 weeks after Clod injection. All cell types were depleted efficiently. Data are pooled from two independent experiments. SD is shown. **e** Cell numbers of spleen macrophages (left), RP-Mps (middle), and DCs (right) normalized to non-injected controls from the same experiments 4 weeks after Clod injection. Macrophages recovered in all mice. RP-Mps failed to recover in all mice. DCs recovered in *Flt3*-proficient but not in *Flt3*-deficient mice. Two-sided *t* tests were performed and SD is shown throughout the figure. SD is shown

that had normal numbers of RP-Mps. In contrast, the regeneration of DCs from LPS-induced depletion was severely blunted in mice that lacked RP-Mps during the regeneration phase (Fig. 6i). We conclude that DC regeneration in adult mice depends on the presence of RP-Mps in the spleen not only under experimental but also under physiological conditions in vivo.

## Discussion
We show here that CSF1R-mediated signals control the DC pool size in FLT3-deficient animals by a cell-extrinsic and non-hematopoietic mechanism providing a novel regulatory pathway to control the differentiation of mature blood cells from adult HSCs. Using novel lineage-tracing mouse tools, we provide evidence that CSF1R is important for the regulation of the DC pool size by an indirect mechanism engaging embryo-derived spleen-resident macrophages that require CSF1R-mediated signals for

their generation during development. With these experiments, we further assign a novel and unprecedented function to spleen tissue-resident RP-Mps of embryonic origin for supporting the establishment and maintenance of a sizable DC pool.

DCs in the spleen of constitutive FLT3 and CSF1R double knockout mice are severely reduced compared to either growth factor receptor mutant alone, defining the growth factors necessary and sufficient for the generation of DCs in vivo. FLT3 is important for homeostatic DC expansion by regulating their division in the spleen[4], whereas, in contrast, we show here that CSF1R-mediated signals are important for the generation of RP-Mps during development that in turn are crucial for the establishment of the DC pool in the spleen of newborn FLT3 mice. Until now, the control mechanisms of CSF1R-mediated signals are exclusively known to be direct and cell intrinsic and instigated by either of the two ligands, CSF1 and IL-34, that provide tissue

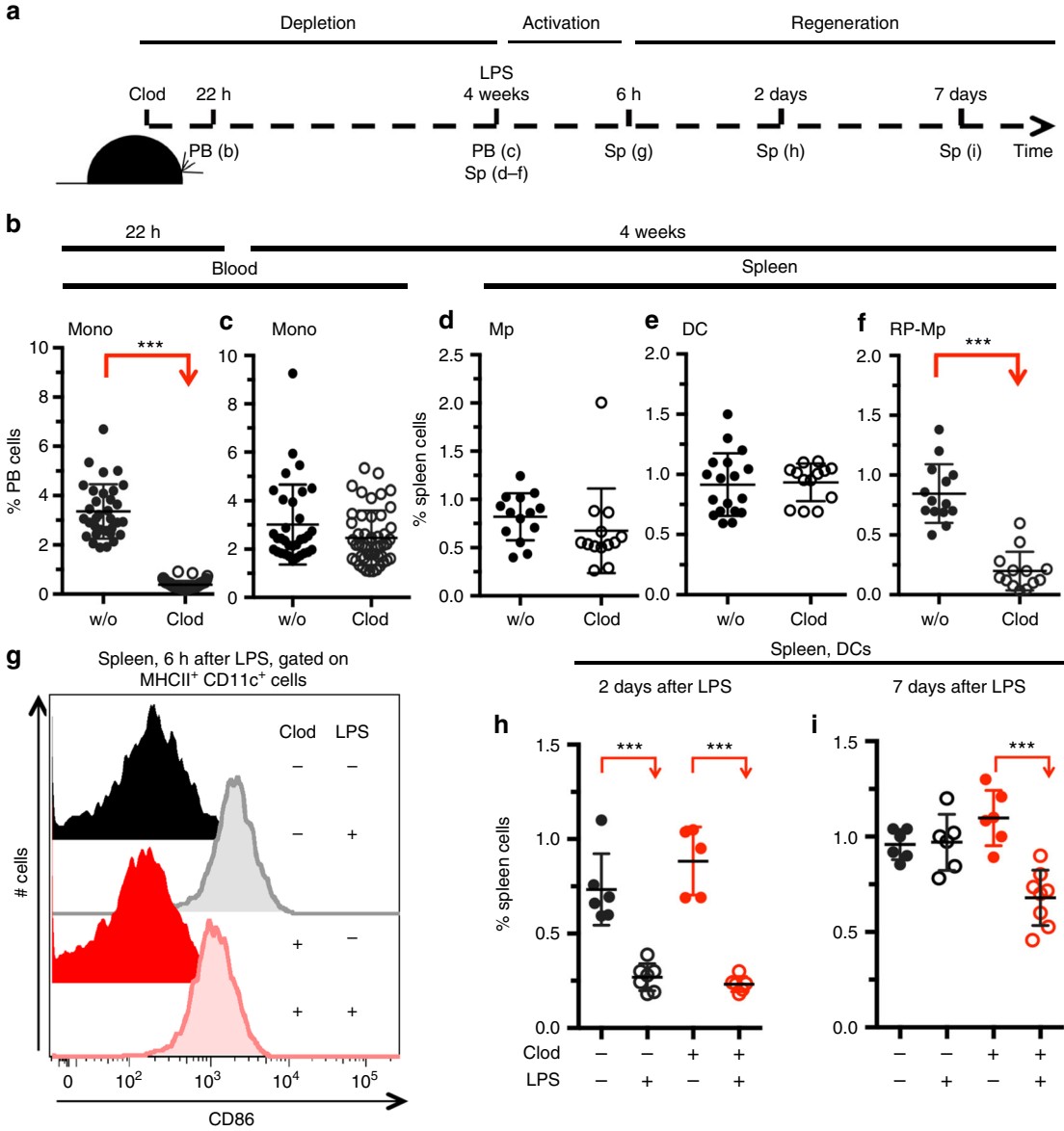

**Fig. 6** RP-Mps in the adult spleen are crucial for the regeneration of DCs after activation-induced depletion in vivo. **a** Schematic outline of the experiment. **b** Plot shows the depletion of blood monocytes 22 h after injection of chlodronate-coated liposomes (Clod). SD is shown. **c** Plot shows blood monocyte regeneration 4 weeks after Clod injection. SD is shown. **d, e** Plots show normal frequencies of spleen macrophages (**d**) and DCs (**e**) 4 weeks after Clod injection. SD is shown. **f** Plot shows reduced frequencies of spleen RP-Mps 4 weeks after Clod injection. SD is shown. **g** Histograms show the expression of CD86 on spleen DCs 6 h after injection of LPS in indicated test groups. **h, i** Plots show the frequencies of DCs in mice that have or have not received Clod 4–5 weeks before and that were injected with LPS 2 (**h**) or 7 (**i**) days before. Data are pooled from three independent experiments. Two-sided *t* tests were performed and SD is shown throughout the figure. SD are shown

specificity to CSF1R-mediated effects on macrophages populations[35]. Consistent with effects of lack of CSF1R[26], CSF1R-mediated signals are important for the generation of embryonic macrophages in the fetal liver during development and in the spleen of neonate mice. Thus spleens of newborn mice deficient for CSF1R exhibit only 15% of the RP-Mp density compared to wild-type mice but the frequencies ameliorate to wild-type levels over time[26]. In turn, the presence of embryo-derived macrophages in the spleen is pivotal for the establishment of the adult HSC-derived DC pool, suggesting that a crosstalk between these cell types is essential for DC differentiation. In adult mice, RP-Mps remain in charge of controlling the DC pool size in the spleen of FLT3 mice but become independent of CSF1R-mediated signals for their generation. The precise mechanism of crosstalk between RP-Mps and DCs remains unknown. However, using an

in vitro differentiation assay, spleen cells enhance the differentiation of DCs in vitro and this support was found independent of direct cell–cell contact, suggesting that a soluble factor may be the molecular mediator of that support. Taken together, our data adds another mechanism of CSF1R-mediated control of myeloid cell differentiation via regulating the generation of RP-Mps.

In *Flt3*-deficient mice, spleen DC numbers are reduced due to a cell-intrinsic requirement for FLT3 for homeostatic DC division[3,4]. *Csf1r*- or *Csf1*-deficient mice have normal[11] or mildly reduced[36] numbers of spleen DCs, respectively, and only the combined deficiency of *Flt3* and *Csf1r* results in the complete absence of spleen DCs in situ. We show here that cell-extrinsic support for DC differentiation becomes evident and important exclusively under stress situations. These stressors can be FLT3 deficiency, which is experimental stress during development

(Fig. 1) and in adult mice (Fig. 5), or the need for rapid regeneration after activation-induced depletion in situ, which provides a physiological challenge in adult mice (Fig. 6). Thus only in the context of the establishment or maintenance of a normally sized DC pool the CSF1R-activated pathway is crucial. The support provided by CSF1R-mediated signals can be seen as a novel kind of redundancy where the effects of the supporting receptor targets a different cell type and thereby works by a cell-extrinsic mechanism. This may serve as a support system to ensure the presence of immune regulatory DCs in situ. Taken together, our data link blood cell differentiation from adult HSCs to a cell type of a different ontogeny, providing a novel regulatory principle for innate immune cell differentiation. The interdependency between cells of different ontogenies may be just an example, and the differentiation of other cells of the mononuclear phagocyte system and may be also of adaptive immune cells may depend on similar principles.

The spleen contains several subpopulations of macrophages. The marginal metallophilic macrophages and marginal zone macrophages are considered important bridges between innate and adaptive immunity, whereas RP-Mps are mainly accounted responsible for iron homeostasis by scavenging senescent erythrocytes[37]. Consistent with this function is the heme-induced expression of Spi-C, a transcription factor that specifically controls the differentiation of RP-Mps during development[38] and from monocytes in adult mice under conditions of pathologic hemolysis[20]. RP-Mps are also thought to contribute to the priming of adaptive immune responses by shaping the progression of inflammatory response after injury or infection and subsequent return to homeostasis[37]. We provide data here that suggests the addition of a very different role for RP-Mps in the regulatory circuit controlling the differentiation of an innate immune cell type from adult HSCs. The generation of the spleen RP-Mp pool during development is crucial for the establishment of the DC pool in neonate FLT3 mice. Moreover, in adult mice, the presence of RP-Mps remains pivotal for the maintenance of the DC pool size because depletion of RP-Mps in $Flt3^{-/-}$ mice results in the loss of DCs. Finally, lack of RP-Mps in wild-type mice impinges on the regeneration of DCs after their activation-induced depletion in situ. RP-Mps are a mixed cell population that initially largely consists of embryo-derived cells that are progressively replaced with age by cells of an identical phenotype but originating from definitive HSCs[15,16]. Our data suggest that RP-Mps of different ontogenetic origin have distinct requirement for CSF1R-mediated signals for their differentiation but fulfill the same function with respect to DC pool size control. We conclude that the spleen microenvironment, specifically RP-Mps, represents a novel regulator for maintaining immune cell integrity within tissues.

Collectively, our results provide evidence that a crosstalk between hematopoietic cell types of distinct origins is required for steady-state hematopoiesis throughout life, implying existence of a novel layer of complexity for the understanding and potentially manipulation of differentiation processes in vivo. In future studies, it will be interesting to determine whether direct or indirect mechanisms account responsible for DC pool size control through RP-Mps and to decipher the molecular nature of DC–RP-Mp interactions in vivo. Moreover, this interdependency between cells of different ontogenies may be just an example, and the differentiation of other cells of the mononuclear phagocytic system and may be also cells of adaptive immunity may depend on similar principles.

## Methods
**Mice**. The following mouse strains were bred and kept under specific pathogen conditions in separated ventilated cages in the animal facility of the TU Dresden

and they were kindly provided by: $Csf1r^{-/-}$[25] from Richard Stanley, $Csf1r^{F/F}$[27] from Jeffrey Pollard, $Flt3^{-/-}$[24] from Ihor Lemischka, $Vav$-$Cre$[29] from Thomas Graf, $Rosa.26$-$CreER^{T2}$ ($R26$-$CreER^{T2}$)[30] from Pierre Chambon and Anton Berns, and $Td$-$rfp$ mice[39] from Jörg Fehling. $CD11c$-$Cre$ (#008068), $Cx3cr1$-$gfp$ (#008451), $Pgk$-$Cre$[32] (#020811), and C57BL/6J (#000664) mouse strains were purchased from the Jackson laboratory. CD-1 lactating females were provided by the Transgenic Core Facility MPI-CBG Dresden. Owing to lack of teeth in $Csf1r^{-/-}$ mice[25], all experiments were performed with mice maximal 3 weeks of age that were kept with the lactating mother to avoid secondary effects from malnutrition.

Generation of Rank-iCre mice: The bacterial artificial chromosome (BAC) containing $Rank$ ($Tnfrsf11a$, clone name: RP24-353D23) was modified by recombineering[40] to introduce an iCre-pA-FRT-PGK-Em7-neo-polyA-FRT cassette into the first exon of $Rank$. The homology arms are 8.9 kb for 5′ and 9.6 kb for 3′. R1 embryonic stem cell were cultured with fetal calf serum (FCS)-based medium [Dulbecco's modified Eagle's medium+GlutaMAX™ (Invitrogen), 15% FCS (PAA), 2 mM L-glutamine (Invitrogen), 1× non-essential amino acids (Invitrogen), 1 mM sodium pyruvate (Invitrogen), 0.1 mM β-mercaptoethanol, in the presence of 1000 units LIF (Chemicon) per ml] on mitomycin-C inactivated mouse embryonic fibroblasts. Cells ($1 \times 10^7$) were electroporated with 40 μg linearized targeting construct using standard conditions (250 V, 500 μF) and selected with 0.2 mg/ml G418. Colonies were screened for correct targeted events by Southern blot hybridization using an internal probe and 5′ and 3′ external probes. Two correctly targeted clones were selected for morula laser injection. The chimera were bred to C57BL/6 mice resulting in germ line transmission for clone 28. RankiCre + /− mice were crossed to $CAGGS$-$Flpo$[41] in order remove the PGK-neo cassette. The mice were subsequently bred back three times to C57BL/6J mice. Animals were assigned to groups based on their genotype and the investigator was blinded to the group allocation for analysis of embryos or newborn mice. Genoytping primers are listed in Table S1. All animal experiments and protocols were approved by the relevant authorities: Landesdirektion Dresden, Saxony, Germany.

Cell isolation from embryos: To analyze embryos, timed pregnancies were performed and the day of embryonic development was estimated by taking the day of vaginal plug as 0.5 days post-conception, termed 0.5. Yolk sac from E10.5 embryos was digested 60 min at 37 °C in 5% FCS/phosphate-buffered saline (PBS) containing 0.3 U/ml collagenase D (Roche) and 100 μg/ml DNAseI (Sigma-Aldrich). The reaction was stopped by incubation with 12.5 mM EDTA. Fetal liver was processed similarly but it was gently disintegrated before digest by pipetting and digested for 20 min.

Tamoxifen: TAM was introduced to adult mice by combination of oral gavage and TAM-containing diet. TAM was dissolved in SMOFlipid oil (Fresenius Kabi) overnight at 4 °C and used the next day. Adult mice were gavaged twice at the age of 8–12 weeks with a 1-week interval with 5 mg of TAM and kept on TAM diet for 6–8 weeks. Pregnant females were induced by a single TAM gavage (2 mg) at E10.5, E14.5, or E17.5 as indicated. To prevent abortions, progesterone (37.5 μg/g body weight resolved in Sunflower seed oil, Sigma-Aldrich) was injected intraperitoneally directly after gavage. To analyze newborn or 3-week-old mice, caesarean sections were carried out at term and neonates were fostered using lactating CD-1 females.

Clodronate liposomes: Clodronate liposomes (clodronateliposomes.org) were injected intravenously (i.v.; 5 μl/g body weight) and $Flt3^{-/-}$ or wild-type mice were bled 24 h later to test for blood macrophages and analyzed 4 weeks later for spleen myeloid cells (Fig. 5). Clodronate liposomes were injected into wild-type mice and depletion was confirmed 22 h later. Regeneration of blood monocytes, spleen macrophages, and DC was confirmed 4 weeks later. In this time interval, RP-Mps had not recovered. LPS (25 μg, i.v., Sigma) was injected into Clod-treated and wild-type mice and the activation of DCs was confirmed. Cell numbers of spleen DCs were analyzed 2 and 7 days after LPS treatment (Fig. 6).

**Flow cytometry**. Spleen and BM cell suspensions were prepared, counted, and stained using the antibodies indicated below[42]. BM cells were harvested by crushing or flushing femurs using a syringe and a 23 G needle with 10 ml of PBS/5% FCS. BM was dissociated by gently pipetting up and down with a 1 ml pipette. Spleens were gently dispersed between frosted slides and digested for 30 min at 37 °C in PBS with 5% FCS containing Collagenase Type 4 (Worthington) at a final concentration of 4.2 U/ml and 100 μg/ml DNAseI (Sigma-Aldrich). The reaction was stopped by incubation with 12.5 mM EDTA. Peritoneal cells were harvested by flushing the peritoneal cavity with 10 ml PBS/5% FCS heated to 37 °C. Spleens of $Csf1r^{-/-}$ and $Csf1^{op/op}$[36] mice are smaller compared to controls. To take overall loss of cellularity into account, the fold change of individual cell types were compared to the fold change of organ cellularity. Fold changes were calculated by dividing each data point of the mutant and the control genotype with the mean of wild-type samples for each experiment. Cells were stained using antibodies specific for the following antigens: B220 (RA3-6B2), CD3 (2C11), CD11b (M1/70), CD11c (N418), CD19 (1D3), CD45 (30-F11), CD45.1 (A20), CD45.2 (104), CD172a (P84), CD117 (2B8), CD115 (AFS98), CD135 (A2F10), F4/80 (BM8), Gr-1 (RB6-8C5), NK1.1 (PK136), Sca-1 (D7), TER119 (Ter119) (eBioscience), MHCII (AF6-120.1, BD Pharmingen), and Streptavidin PB (Molecular Probes). There are no differences in spleen DC numbers between wild-type and heterozygous $Flt3$ or $Csf1r$ mice or $Cre^+;Flt3^{F/+}$ and $Cre^+;Flt3^{+/+}$ or $Cre^+;Csf1r^{F/+}$ and $Cre^+;Csf1r^{+/+}$ mice,

thus cell numbers from these control mice were pooled throughout the manuscript. Samples were acquired and sorted using an LSRII and AriaII cytometer (BD), respectively, and the data were analyzed using the FlowJo software (Tree Star).

**Immunohistology.** Spleens were frozen in OCT embedding medium (Tissue-Tek) according to the manufacturer's protocol and 3-μm-thick cryosections were performed on Cryotome CM1900 (Leica). Sections were fixed in acetone and blocked with rat Ig (500 μg/ml, Jackson ImmunoResearch Laboratories) and the Streptavidin/Biotin Blocking Kit (Vector Laboratories). Samples were stained with antibodies directed against the following antigens: F4/80 (A3-1, Abcam), rabbit anti-RFP (polyclonal, Rockland), CD11b (M1/70), CD11c (N418), CD3 (2C11), and B220 (RA3-6B2) (all from eBioscience). Anti-FITC-Alexa488 (Molecular Probes), anti-rabbit IgG A555 (Life Technologies), anti-rat IgG Cy5 (Jackson), and streptavidin-Cy3 (Jackson ImmunoResearch Laboratories) were used for secondary steps. Dapi (4,6-diamidino-2-phenylindole; 2 μg/ml) was used to stain nuclei. Stained sections were mounted with fluoromount G (Southern Biotech). Samples were analyzed by fluorescence microscope BZ-9000E (Keyence) and images were processed by the Fuji software.

**Molecular analysis.** Genomic DNA from sorted cells, spleen, BM, and tail was isolated by using the DNeasy® Kit (Qiagen) according to the manufacturer's protocol. Following primers were used to detect the $Csf1r^F$ and $Csf1r^{delta}$ alleles: fwd, ATCCTCAAACGTGGAGACACC; rev,GCCACCATGTCTCCGTGCTT. PCR specific for the glycerol-3-phosphate dehydrogenase (Gapdh) gene locus was used as loading control: fwd, TACGCATTATGCCCGAGGAC; rev, TGTAGGCCAGGTGATGCAAG.

**Statistics.** Student's unpaired, two-sided $t$ test was performed to calculate the statistical significance between individual groups with expected normal distribution. Mann–Whitney $U$ test was performed to calculate the statistical significance between groups where normal distribution was not expected (fold-change comparisons). Not significant (ns) = $P$ value $(P) > 0.05$; *$P = 0.05$–$0.01$; **$P = 0.01$–$0.001$; ***$P < 0.001$. Prism 5 software (GraphPad) was used to perform statistical analysis. If not indicated differently, mean and standard deviation is shown. The program G*Power 3.1 was used to estimate sample size.

## Data availability

All data generated or analyzed during this study are included in this published article (and its supplementary information files). Primary data files are available from the corresponding author on reasonable request.

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

## Acknowledgements

The authors are grateful to Joerg Fehling, E. Richard Stanley, and Thomas Graf for providing *Td-rfp*, *Csf1r*-null and *Vav-Cre* mice, respectively. We thank Melanie Portz for expert technical assistance. This work was supported by the German Research Foundation (DFG) through WA2837/1-1, FOR2033-A03, TRR127-A5, WA2837/6-1, and WA2837/7-1 to C.W. and Wellcome Trust (101067/Z/13/Z) to J.W.P.

## Author contributions

G.I.P. and J.E. designed and performed experiments, analyzed data, and made the figures, J.W.P. provided crucial help. J.F., A.K., and R.N. generated Tnfrsf11a-iCre knock-in mouse model, and R.N. advised on plug breedings, TAM injections, and fostering. C.W. conceived the study, designed experiments, analyzed data, and wrote the manuscript.

## Additional information

**Competing interests:** The authors declare no competing interests.

