## [Peer Review File · Nature Communications]

CSF1R regulates the dendritic cell pool size in adult mice via embryo-derived tissue-resident macrophages.

Percin et al.

Supplementary Figure 1

Supplementary Fig. 1. Identification and numbers of spleen cells and their precursors. (a) Spleen cells were analyzed for the cell surface expression of CD11b, Gr-1, MHCII, CD11c, F4/80 as indicated to identify T and B lymphocytes, neutrophils (PMN), DCs, red-pulp macrophages (RP-Mp), eosinophils, and monocytes (top). Labeling frequencies of indicated cell types using *Rank-iCre⁺;eYFP^{wt/KI}* mice (bottom two rows). (b) Spleen cellularity (left) and DC numbers (right) in mice of indicated genotypes. SD is shown. (c) Dot plots show spleen DCs (MHCII⁺ CD11c^{hi}) from mice of indicated genotypes that were resolved for the expression of CD8 and CD11b (left). Frequencies of spleen DC subsets (middle). Fold-change of spleen cells and DC subsets of indicated genotypes. Data is pooled from 6 experiments (right). SD is shown. (d) Spleen macrophage numbers (CD11b⁺ F4/80^{lo}) in mice of indicated genotypes. SD is shown. (e) Quantification of DCs on histological sections shown in Fig. 1e of the manuscript. DC numbers were determined per vision field (2-4 counted) using sectioned samples from mice from 2 independent experiments. SD is shown. (f) Frequencies, numbers, and fold-change of spleen DCs of indicated genotypes in C57BL/6J congenic mice. The null allele of *Csf1r* was bred back 10-times onto the C57BL/6J genetic background; the viability of *Csf1r^{-/-}* mice decreased to 0.87%. We analyzed a total of 578 mice from heterozygous breedings and obtained 5 double mutant mice over a time period of four years. SD is shown. (g) Fold-change of bone marrow (BM) cells and MDPs of indicated genotypes. MDP: Lin⁻ (Lin=CD3⁺ CD19⁺ NK1.1⁺ Ter119⁺ CD11b⁺ Gr-1⁺ B220⁺) Sca-1⁻ CD115⁺ ¹. SD is shown. (h) Identification of pre-cDC1 (Lin⁻ MHCII⁻ CD11c^{hi} CD172a⁻ Ly6C⁻ Siglec-H/CD33⁻) and pre-cDC2 (Lin⁻ MHCII⁻ CD11c^{hi} CD172a⁻ Ly6C⁺ Siglec-H⁺) ^{2,3} in the bone marrow (left) and spleen (right) in constitutive (top) and conditional (bottom) *Flt3* and *Csf1r* null mice. (i,j) Quantification and fold-change of pre-cDC1 and pre-cDC2 cells in mice of indicated genotypes in the bone marrow (i) or spleen (j). SD are shown.

Supplementary Figure 2

Supplementary Fig. 2. Efficient depletion of CSF1R expression using the LoxP-flanked *Csf1r* allele. (a) *CD11c-Cre⁺;tdRFP^{wt/ki}* mice ⁴ were generated and indicated cell types identified by cell surface staining and the frequency of tdRFP⁺ cells within these populations determined (open circles *CD11c-Cre⁺;tdRFP^{wt/ki}*; closed circles *CD11c-Cre⁺;tdRFP^{wt/wt}*). Pre-cDC were identified as Lin⁻ (Lin=CD3 CD19 Ter119 NK1.1 B220) MHCII⁻ CD11c^{hi} Flt3⁺ Sirpa^{lo} cells ⁵. SD is shown. (b,c) Recombination efficiency in indicated sorter-purified cells from *CD11c-Cre⁺;Flt3^{-/-};Csf1r^{F/-}* (b) or *Vav-cre⁺;Flt3^{-/-};Csf1r^{F/-}* (c) mice. (d) Scheme of TAM induction using adult *R26-CreER^{T2+}* deleter mice. (e,f) Dot plots show the expression of Gr-1 and CSF1R on CD11b⁺ SSC^{lo} blood monocytes one week after TAM induction in *R26-CreER^{T2+};Flt3^{-/-};Csf1r^{F/-}* and control mice. Graph summarizes the expression of CSF1R on blood monocytes pooled from 4 independent experiments that were TAM-induced 1-2 weeks before (f). SD is shown. (g) Dot plots show the expression of KIT and CSF1R on BM progenitor cells (Lin⁻ [Lin=CD3 CD19 NK1.1 Ter119 CD11b Gr-1 B220] Sca-1⁻) from *R26-CreER^{T2+};Flt3^{-/-};Csf1r^{F/-}* and control mice 6-8 weeks after TAM induction. Data is pooled from 4 independent experiments. SD is shown. (h) Plots show cell surface expression of CSF1R on spleen CD11b⁺ F4/80^{lo} macrophages in adult *R.26-CreER^{T2+};Flt3^{-/-};Csf1r^{F/-}* mice that were TAM-treated 6-8 weeks before. CSF1R expression was normalized to TAM-treated *R.26-CreER^{T2+};Flt3^{-/-};Csf1r^{F/+}* controls. Data shown is pooled from 4 independent experiments. SD is shown. (i) Recombination efficiency in total spleen or bone marrow cells from *R26-CreER^{T2+};Flt3^{-/-};Csf1r^{F/-}* mice that were TAM-induced 6-8 weeks earlier at 8-12 weeks of age. (j) Cell numbers of large peritoneal macrophages (left) and spleen RP-MP (right) in adult *R.26-CreER^{T2+};Flt3^{-/-};Csf1r^{F/-}* mice that were TAM-treated 6-8 weeks before. Data is pooled from two (PEC) and 7 (spleen) independent experiments. SD is shown. (k) Recombination efficiency determined using tail gDNA from *Pgk-Cre⁺;Csf1r^{F/-}* and control mice. (l) Dot plots show the expression of KIT and CSF1R on Lin⁻ Sca-1⁻ bone marrow cells of *R26-creER^{T2+};Flt3^{-/-};Csf1r^{F/-}* and control mice that were TAM-induced at E10.5 and analyzed at 3 weeks of age. Quantification of CSF1R expression on Lin⁻ Sca-1⁻ bone marrow cells (right). Data from 3 experiments was pooled. SD is shown.

Supplementary Figure 3

Supplementary Fig. 3. Generation and lineage tracing using *Rank-iCre*⁺ mice. (a) Southern blot analysis reveals correct insertion of the construct into the *Rank* gene. **(b-d)** Identification of cell populations depicted in **Fig. 4c**.

Table S1. List of all primers used in the manuscript.

Allele	Fwd primer (5'  3')	Rev primer (5'  3')
Rank-iCre+	AACCTGAGGATGTGAGGGACTA	GTCAAAGTCAGTGC GTTCAAAG
CD11c-Cre+	GCCTGCATTACCGGTTCGATGCAACGA	GTGGCAGATGGCGCGGCAACACCATT
Vav-Cre+	GCCTGCATTACCGGTTCGATGCAACGA	GTGGCAGATGGCGCGGCAACACCATT
Pgk-Cre+	GCCTGCATTACCGGTTCGATGCAACGA	GTGGCAGATGGCGCGGCAACACCATT
R26-CreERT2+	GCCTGCATTACCGGTTCGATGCAACGA	GTGGCAGATGGCGCGGCAACACCATT
CX3CR1-GFP+	CTTCTTCAAGGACGACGGCAACTA	ATCGCGCTTCTCGTTGGGGTCTTTGC
Csf1r wt	TCTCCTGGGATGGGAAACGATCCCAA GGC	GATTCAGGGTCCAAGGTCCAGATGGGAG AG
Csf1r -	GGTGGATGTGGAATGTGTGCG	CGTTTCTTGTGGTCAGGGTGC
Csf1r F	ATCCTCAAACGTGGAGACACC	GCCACCATGTCTCCGTGCTT
Csf1r del	CAGATGCTAGCCCTGTGATGG	CTTCAAGCTGCAGCCCAAACCTC
Flt3 wt	TCCACGTTGTTCCCTCTACC	TATGTGGGCAATTTGGCTCT
Flt3 -	TGATCTCGTCGTGACCCAT	TATGTGGGCAATTTGGCTCT

Supplementary References

- 1 Waskow, C. *et al.* The receptor tyrosine kinase Flt3 is required for dendritic cell development in peripheral lymphoid tissues. *Nat Immunol* **9**, 676-683 (2008).
- 2 Schlitzer, A. *et al.* Identification of cDC1- and cDC2-committed DC progenitors reveals early lineage priming at the common DC progenitor stage in the bone marrow. *Nat Immunol* **16**, 718-728 (2015).
- 3 Sichien, D. *et al.* IRF8 Transcription Factor Controls Survival and Function of Terminally Differentiated Conventional and Plasmacytoid Dendritic Cells, Respectively. *Immunity* **45**, 626-640 (2016).
- 4 Luche, H., Weber, O., Nageswara Rao, T., Blum, C. & Fehling, H. J. Faithful activation of an extra-bright red fluorescent protein in "knock-in" Cre-reporter mice ideally suited for lineage tracing studies. *Eur J Immunol* **37**, 43-53 (2007).
- 5 Liu, K. *et al.* In vivo analysis of dendritic cell development and homeostasis. *Science* **324**, 392-397 (2009).